A comparative analysis of the complete chloroplast genomes of three Chrysanthemum boreale strains

Tyagi Swati 1
Jung Jae-A 2
http://orcid.org/0000-0001-5209-4481 Kim Jung Sun 1
http://orcid.org/0000-0001-5015-3150 Won So Youn 1 soyounwon@korea.kr
1 Genomics Division, National Institute of Agricultural Sciences, Rural Development Administration , Jeonju , Republic of Korea
2 Floriculture Research Division, National Institute of Horticultural and Herbal Science, Rural Development Administration , Wanju , Republic of Korea
Silva Pedro
Electronic publication date: 2020 Jul 3
Publication date: 2020
Volume: 8
Electronic Location ID: e9448
Received 2020 Mar 13; Accepted 2020 Jun 9
Copyright: © 2020 Tyagi et al.
Copyright year: 2020
Copyright holder: Tyagi et al.
License: This is an open access article distributed under the terms of the Creative Commons Attribution License, which permits unrestricted use, distribution, reproduction and adaptation in any medium and for any purpose provided that it is properly attributed. For attribution, the original author(s), title, publication source (PeerJ) and either DOI or URL of the article must be cited.
License URL: https://creativecommons.org/licenses/by/4.0/

Keywords: Asteraceae, Anthemideae, Chrysanthemum, Chloroplast genome, Phylogeny, Next generation sequencing

Funding: Cooperative Research Program for Agriculture Science & Technology Development PJ01035802 National Institute of Agricultural Sciences PJ01335301 Rural Development Administration, Republic of Korea This work was supported by the Cooperative Research Program for Agriculture Science & Technology Development (PJ01035802) and a grant from the National Institute of Agricultural Sciences (PJ01335301), Rural Development Administration, Republic of Korea. The funders had no role in study design, data collection and analysis, decision to publish, or preparation of the manuscript.

==============================
Background

Chrysanthemum boreale Makino (Anthemideae, Asteraceae) is a plant of economic, ornamental and medicinal importance. We characterized and compared the chloroplast genomes of three C. boreale strains. These were collected from different geographic regions of Korea and varied in floral morphology.

Methods

The chloroplast genomes were obtained by next-generation sequencing techniques, assembled de novo, annotated, and compared with one another. Phylogenetic analysis placed them within the Anthemideae tribe.

Results

The sizes of the complete chloroplast genomes of the C. boreale strains were 151,012 bp (strain 121002), 151,098 bp (strain IT232531) and 151,010 bp (strain IT301358). Each genome contained 80 unique protein-coding genes, 4 rRNA genes and 29 tRNA genes. Comparative analyses revealed a high degree of conservation in the overall sequence, gene content, gene order and GC content among the strains. We identified 298 single nucleotide polymorphisms (SNPs) and 106 insertions/deletions (indels) in the chloroplast genomes. These variations were more abundant in non-coding regions than in coding regions. Long dispersed repeats and simple sequence repeats were present in both coding and noncoding regions, with greater frequency in the latter. Regardless of their location, these repeats can be used for molecular marker development. Phylogenetic analysis revealed the evolutionary relationship of the species in the Anthemideae tribe. The three complete chloroplast genomes will be valuable genetic resources for studying the population genetics and evolutionary relationships of Asteraceae species.

Introduction

The genus Chrysanthemum belongs to the largest Angiosperm family, the Asteraceae (Hirakawa et al., 2019). Chrysanthemum species are economically important (Hirakawa et al., 2019). They are valued as cut flowers or potted garden flowers due to the diversity of their morphological traits including color, shape and size of the flower head, ray florets and disc florets (Shinoyama et al., 2012; Song et al., 2018). In addition, they are used as medicinal herbs in Korean and Chinese folk medicine (Won, Jung & Kim, 2018) for the treatment of inflammation, asthma and diarrhea, and as a traditional health food (Han et al., 2019; Sun et al., 2015; Wang et al., 2015). Polyploidy and hybridization events were reported to be responsible for evolution and speciation of Chrysanthemum genus (Liu et al., 2012; Ma et al., 2016; Yang et al., 2006), and Chrysanthemum species exhibit diverse ploidy levels (2n = 2x =18 to 2n = 10x = 90) (Chen et al., 2008). The commercial cultivar Chrysanthemum × morifolium Ramat. is a hexaploid species and its genetic studies on important traits and breedings are difficult.

Chrysanthemum includes around 40 different species native to Eurasia, especially in Korea, China and Japan (Liu et al., 2012). However, some species and varieties are narrowly distributed in specific habitats (Kondo et al., 2003; Liu et al., 2012). A total of 8 species, nine subspecies and one variety were reported in Korea (Hoang et al., 2020; Lee, 2006). Of particular importance to the present study is a wild relative, Chrysanthemum boreale Makino, which is a diploid species, bears small yellow flowers, and occurs in natural stands in eastern Asia (Hwang et al., 2013; Kim et al., 2014). Comparative transcriptomic analysis revealed that C. boreale diverged from C. morifolium about 1.7 million years ago (Won et al., 2017). C. boreale is resistant to one of the most destructive fungal diseases, namely white rust caused by Puccinia horiana Henn. (Park et al., 2014), and it has anti-inflammatory and skin-regenerative properties (Kim et al., 2015b, 2010). Several C. boreale strains collected from natural stands in Korea displayed variations in morphology such as leaf shapes and flower head, and in karyotype with the occurrence of aneuploidy (Hoang et al., 2020; Hwang et al., 2013). However, their genetic sequence divergence remains unknown. Currently, work is underway to sequence the nuclear genome of one C. boreale strain aiming to facilitate molecular, genetic, and physiological studies on Chrysanthemum. Molecular markers derived from both nuclear and chloroplast (cp) genomes would help reveal the relationships among strains and the genetic position of C. boreale in Asteraceae.

The cp genome encodes proteins that are key to photosynthesis and other metabolic processes (Liu et al., 2018b). The uni-parental inheritance of the cp genome (usually maternal in angiosperms and paternal in gymnosperms) and conserved gene content and order has made cp genome a valuable asset for plant phylogenetic and evolutionary studies (Birky, 2001; Wu & Ge, 2012). Plant cp genomes are generally between 120 kb and 160 kb in length and have a quadripartite circular structure comprising a pair of inverted repeat (IR) regions, a large single copy (LSC) region, and a small single copy (SSC) region (Thode & Lohmann, 2019). Advances in next-generation sequencing techniques have made it much easier to reconstruct the complete cp genome and uncover phylogenetic relationships at various taxonomic levels (Jansen et al., 2007; Moore et al., 2010; Parks, Cronn & Liston, 2009). Although the structure of cp genome is generally conserved, variation between species, subspecies, and individuals is present, and includes SNPs, indels, sequence rearrangements, IR expansion, gene loss and intron retention (Li et al., 2018). The cp genome sequences have helped to elucidate the phylogenetic relationships and evolutionary history of many plant species, including rice (Oryza AA genome), vegetables in the Brassica genus, and conifer tree (Pinus taeda L.) (Asaf et al., 2018; Kim et al., 2018, 2015a).

Here, we analyzed the cp genomes of three morphologically different C. boreale strains collected from different geographic regions in Korea. We discovered their phylogenetic relationships to other species in the tribe Anthemideae, including Chrysanthemum species. This study provides useful genomic information for molecular evolutionary and phylogenetic studies of Asteraceae, and genetic resources for breeding and improvement of chrysanthemum.

Materials and Methods

Ethics statement

The plant sample used in this study is neither endangered nor protected, and was collected from an area that was not privately owned or protected in any way. No specific permits were required to conduct this study.

Plant materials and sequencing

Two C. boreale strains with morphological differences were collected from different locations (Fig. S1) in the Republic of Korea and deposited at the National Agrobiodiversity Center, Rural Development Administration. The strain from Gongju-si, Chungcheongnam-do was labeled IT232531, and the one from Suwon-si, Gyeonggi-do was labeled IT301358. The total DNA was isolated from fresh leaves as previously described (Kim et al., 2006). The quality and quantity of DNA were examined using a Nanodrop 2000 spectrophotometer (Thermo Fisher Scientific, Waltham, MA, USA) and gel electrophoresis (in 0.8% agarose). Paired-end libraries of 350-bp insert size were constructed using TruSeq DNA PCR-Free kit (Illumina, San Diego, CA, USA) and sequenced with a 101-bp read length by Macrogen (Republic of Korea) using the HiSeq4000 (Illumina, San Diego, CA, USA) according to the manufacturer’s instructions. Another C. boreale strain, labeled 121002, was collected from Jeongeup-si, Jeollabuk-do (Hwang et al., 2013) and its cp genome was sequenced. Our group had previously submitted this cp genome sequence to NCBI with accession number MG913594 (Won, Jung & Kim, 2018).

Chloroplast genome assembly and annotation

The complete cp genome was assembled de novo (Kim et al., 2015a). Briefly, raw reads were trimmed using the Trimmomatic program (Bolger, Lohse & Usadel, 2014), assembled using the clc_assembler in the CLC Genomics Workbench v6.0 (CLC Bio, Denmark, Europe). Gaps were filled using Gap Closer (Luo et al., 2012). The resulting contigs were searched for cp-encoding contigs by BLASTN analysis against the cp genome of C. boreale strain 121002, and circularized. These were annotated using the online programs Dual Organellar GenoMe Annotator, cpGAVAS v.2.0 and BLAST (Shi et al., 2019; Wyman, Jansen & Boore, 2004). The structure of transfer RNA (tRNA) was predicted using the tRNAscan-SE 1.21 program using the default settings (Schattner, Brooks & Lowe, 2005). The circular genome map with structural features was generated using the OGDRAW v1.2 program (Lohse et al., 2013). The resulting cp genome sequences of strains IT232531 and IT301358 were deposited in NCBI under the IDs MN909052 and MN913565, respectively.

Chloroplast genome comparison

The cp genomes of the three C. boreale strains were compared using the mVISTA program in the Shuffle-LAGAN mode, using the annotation of strain 121002 as the reference (Frazer et al., 2004). The SNPs and indels in the cp genome were also recorded using DnaSP6.0 (Rozas et al., 2017) and manually verified from the sequence alignment by Clustal Omega (Sievers et al., 2011).

Characterization of repetitive sequences

Simple sequence repeats (SSRs) were discovered using the online web tool MISA (http://pgrc.ipk-gatersleben.de/misa/) with the following parameters: ten repetitions for mononucleotide motifs, eight for dinucleotide motifs, four for tri- and tetra-nucleotide motifs, and three for penta-and hexa-nucleotide motifs (Beier et al., 2017). Next, four different types of repeats, namely forward (F), palindromic (P), reverse (R) and complement (C) repeats were analyzed using the REPuter program (https://bibiserv.cebitec.uni-bielefeld.de/reputer) with a minimum repeat size of 30 bp and a Hamming distance of 3 (Kurtz et al., 2001). To reduce redundancy, IRb sequence was removed before analysis and repeats detected at the same position were merged into single repeat.

Phylogenetic analysis

The entire cp genomes and 77 protein-coding sequences shared in cp genomes of species belonging to tribe Anthemideae were used to reconstruct the phylogenetic relationships. Lactuca sativa L. was used as the outgroup. The species and the accession numbers of their cp genomes in NCBI are listed in Table S1. The nucleotide sequences were aligned using Clustal Omega (Sievers et al., 2011). Maximum likelihood (ML) analyses were conducted using the IQ-TREE web server (http://iqtree.cibiv.univie.ac.at) with the best-fit models determined by ModelFinder in the IQ-TREE package (Table S2) and 1,000 bootstrap replicates (Hoang et al., 2018; Kalyaanamoorthy et al., 2017; Nguyen et al., 2015). Bayesian inferences (BI) were performed with MrBayes v. 3.2.7 (Ronquist et al., 2012) and the nucleotide substitution models determined by ModelTest-NG (Darriba et al., 2019) (Table S2). The Markov chain Monte Carlo algorithms were run for 10 million generations and sampled every 1,000 generations. The first 25% of trees were discarded as burn-in and the remaining trees were used to build a majority-rule consensus tree with posterior probability values for each node. The stationary was considered to be reached when the average standard deviation of split frequencies remained below 0.01. The phylogenetic tree was visualized with FigTree v1.4.4 (http://tree.bio.ed.ac.uk/software/figtree/).

Results

Characterization of chloroplast genomes

We used NGS techniques to generate approximately 30.2 Gb and 34.9 Gb of raw reads from strains IT232531 and IT301358, respectively. We assembled de novo the complete cp genomes of sizes 151,098 bp for IT232531 and 151,010 bp for IT301358 (Table 1). For comparison, we included the previously reported cp genome of C. boreale strain 121002, which was 151,012 bp in size (Won, Jung & Kim, 2018). All the three C. boreale strains had a typical quadripartite structure of cp genomes with an LSC, an SSC, and a pair of IR regions (Fig. 1). The length of the LSC region was 82,817 bp, 82,880 bp and 82,788 bp for the strains 121002, IT232531 and IT301358, respectively. The SSC region measured 18,281 bp, 18,312 bp and 18,310 bp in the strains 121002, IT232531 and IT301358, respectively. The strains were comparable in terms of the length of the IR regions and the GC content of the LSC, SSC, IR regions and the complete genome (Table 1). The IR regions had a higher GC content than the LSC and SSC regions due to the presence of GC-rich ribosomal RNA (rRNA) genes and tRNA genes in these regions.

Figure 1 Genome map of Chrysanthemum boreale chloroplast genomes.

Thick lines indicate the extent of the inverted repeat regions, which separate the genome into large and small single copy regions. Genes drawn inside the circle are transcribed clockwise, while those outside of the circle are transcribed counter clockwise. Genes belonging to different functional groups are color coded differently. The dark gray in the inner circle corresponds to the GC content while the light gray corresponds to the AT content. Genes with introns are marked with an asterisk.

Table 1 Summary of complete chloroplast genomes of three Chrysanthemum boreale strains.

Attributes	121002	IT232531	IT301358	
Total size (bp)	151,012	151,098	151,010	
LSC size (bp)	82,817	82,880	82,788	
SSC size (bp)	18,281	18,312	18,310	
IR size (bp)	24,957	24,953	24,956	
Total GC content (%)	37.5	37.5	37.5	
LSC GC content (%)	35.6	35.5	35.6	
SSC GC content (%)	30.9	30.8	30.9	
IR GC content (%)	43.1	43.1	43.1	
Number of unique genes	113	113	113	
Number of unique protein-coding genes	80	80	80	
Number of unique tRNA genes	29	29	29	
Number of unique rRNA genes	4	4	4	
Genes duplicated	17	17	17	
Genes with intron	16	16	16	
Pseudogene	1	1	1	

The cp genomes of all the strains comprised 113 unique genes. These included 80 protein-coding genes, 29 tRNA genes and four rRNA genes (Table 2). Each strain contained 61 protein-coding genes and 21 tRNA genes in the LSC region and 11 protein-coding genes and one tRNA gene in the SSC region (Fig. 1). Three genes (rps12, rps19 and ycf1) were distributed in both single copy and IR regions. The IR regions contained seven protein-coding genes, seven tRNA genes and four rRNA genes each. Because the IR regions are duplicates of each other, all genes in these regions were also duplicated.

Table 2 List of genes in the C. boreale chloroplast genomes.

Category	Group of genes	Name of genes	
Self-replication	Large subunit of ribosomal proteins	rpl2*(2x), 14, 16, 20, 22, 23(2x), 32, 33, 36	
Small subunit of ribosomal proteins	rps2, 3, 4, 7(2x), 8, 11, 12**(2x), 14, 15, 16*, 18, 19	
DNA dependent RNA polymerase	rpoA, B, C1*, C2	
rRNA genes	rrn16(2x), rrn23(2x), rrn4.5(2x), rrn5(2x)	
tRNA genes	trnA-UGC*(2x), trnC-GCA, trnD-GUC, trnE-UUC, trnF-GAA, trnfM-CAU, trnG-GCC, trnH-GUG, trnI-CAU(2x), trnI-GAU*(2x), trnK-UUU*, trnL-CAA(2x), trnL-UAA*, trnL-UAG, trnM-CAU, trnN-GUU(2x), trnP-UGG, trnQ-UUG, trnR-ACG(2x), trnR-UCU, trnS-GCU, trnS-GGA, trnS-UGA, trnT-GGU, trnT-UGU, trnV-GAC(2x), trnV-UAC*, trnW-CCA, trnY-GUA	
Photosynthesis	Photosystem I	psaA, B, C, I, J	
Photosystem II	psbA, B, C, D, E, F, H, I, J, K, L, M, N, T, Z	
NADH dehydrogenase	ndhA*, B*(2x), C, D, E, F, G, H, I, J, K	
Cytochrome b6/f complex	petA, B*, D*, G, L, N	
ATP synthase	atpA, B, E, F*, H, I	
Rubisco	rbcL	
Other genes	Translational initiation factor	infA	
Maturase	matK	
Protease	clpP*	
Envelop membrane protein	cemA	
Subunit Acetyl-CoA-Carboxylase	accD	
C type cytochrome synthesis gene	ccsA	
Unknown	Conserved open reading frame	ycf1, 2(2x), 3*, 4, 15(2x)	
Notes:

* Intron-containing genes.

** Trans-spliced gene.

The duplicated genes are shown with (2x) next to the gene name.

The cp genomes of C. boreale included 16 intron-containing genes (Table 3). The genes ycf3 and clpP had two introns each, while all other genes contained a single intron. Nine of the introns were identical in length, whereas seven other introns differed in length between 1 bp and 24 bp. The intron of the trnK-UUU gene was largest (2,560–2,575 bp) in all the strains and its pairwise length differed between the strains by 7–15 bp. The intron of the ndhA gene in IT232531 was 24 bp and 6 bp longer than that in strains 121002 and IT301358, respectively. In each strain, the rps12 gene was trans-spliced, with the 5′ end exon located in the LSC region and the duplicated 3′ end exon located in both the IR regions, as previously reported in other plants (Thode & Lohmann, 2019).

Table 3 Comparison of introns length of C. boreale strains in cp genome.

No.	Genes	Location		121002	IT232531	IT301358	
1	atpF	LSC		699	699	699	
2	clpP	LSC	Intron1	608	609	611	
			Intron2	800	797	797	
3	ndhA	SSC		1045	1069	1063	
4	ndhB	IR		670	670	670	
5	petB	LSC		747	746	747	
6	petD	LSC		675	675	675	
7	rpl2	LSC		662	662	662	
8	rpoC1	LSC		732	732	732	
9	rps12	IR		535	535	535	
10	rps16	LSC		881	892	887	
11	ycf3	LSC	Intron1	740	743	743	
			Intron2	711	713	711	
12	trnA-UGC	IR		812	812	812	
13	trnI-GAU	IR		776	776	776	
14	trnK-UUU	LSC		2568	2575	2560	
15	trnL-UAA	LSC		424	423	425	
16	trnV-UAC	LSC		572	572	572	

Given that the cp genome of C. boreale strain 121002 was obtained using PacBio’s long reads (Won, Jung & Kim, 2018), we repeated the cp genome assembly of 121002 using Illumina’s short reads as conducted for other C. boreale strains. The sequence comparison between two cp genomes obtained using long reads and short reads revealed that there was no SNP detected. Instead, indels were observed at four genomic regions and all of them were associated with homopolymers. Three indels were located in intergenic spacers (IGSs), trnE-UUC_rpoB (18 thymines in the long-read assemble vs. 17 thymines in the short-read assemble) and psaA_ycf3 (16 vs. 15 adenines), and the intron of rpl16 (8 vs. 9 cytosines), which didn’t change the protein sequences. However, the coding region of ycf1 possessed one indel (13 adenines vs. 14 adenines) (Fig. S2), which resulted in 1,036 amino acids (aa) in the original data due to the premature stop codon. However, one-bp insertion generated the ycf1 protein of 1,668 aa, which was more consistent with the other C. boreale strains (1,672 aa in IT232531 and 1,673 aa in IT301358). While we used the original cp sequence deposited in NCBI for analyses, in case of ycf1, we used the newly obtained sequences.

Variation in chloroplast genomes

The mVISTA-based identity plot indicated conservation in DNA sequence and gene synteny across the whole cp genome, and revealed the regions with increased genetic variation (Fig. 2). The gene number, order and orientation were conserved. There was higher genetic variability in the single copy (LSC and SSC) regions than in the IR regions, and in non-coding regions than in coding regions. Highly diverged regions included the IGSs, trnK-UUU_rps16, trnS-GCU_trnC-GCA, trnR-UCU_trnT-GGU, rps4_trnL-UAA, ndhC_trnV-UAC, psbE_petL, rps16_rps3, and trnL-UAG_rpl32 and the introns of trnK-UUU, rps16 and ndhA (Fig. 2). We detected a total of 298 SNPs (Table S3). The LSC region contained a majority of the SNPs (204, accounting for 68.5% of the SNPs), followed by the SSC region (75, 25.2%), and the IR regions (19, 6.4%). The SNPs were more abundant in non-coding regions: 141 were located in intergenic regions, 46 in introns, and 111 in coding regions. The ycf1 gene contained the largest number of substitutions (25 SNPs), followed by the trnK-UUU intron (18 SNPs), rpoC2 (12 SNPs) and the ycf1_rps15 IGS (11 SNPs).

Figure 2 Comparison of chloroplast genomes of C. boreale strains using the mVISTA program.

A cut-off of 70% identity was used for the plots. The Y-scale axis represents the percent identity between 50% and 100%.

We detected a total of 106 indels (Table S4): 81 in the LSC, 19 in the SSC, and six in the IR regions. A total of 86 and 17 indels were located in IGS and introns, respectively, whereas three were contained in coding regions. The ndhC_trnV-UAC spacer had five indels, while the introns of trnK-UUU and ndhA, and the spacers of psaA_ycf3 and psbE_petL contained four indels each. The psbE_petL IGS included the two largest indels (54 bp and 36 bp) in the cp genome. The trnK-UUU intron was the longest in the genome, and one of the most variable regions, comprising both SNPs and indels (Fig. S3). The 5-bp deletion at the end of the protein-coding gene rpoC2 in strain IT232531 generated a protein that was longer by two amino-acids. In the ycf1 gene, the 3-bp insertion in strain IT301358 did not change the protein’s translational frame.

We investigated the position of genes at the junction regions (LSC/IRa, IRa/SSC, SSC/IRb and IRb/LSC; Fig. 3). At the LSC/IRa junction, C. boreale possessed rps19 with 220 bp in LSC and 59 bp in IRa. The IRa/SSC junction contained the functional ycf1, while the SSC/IRb possessed the duplicated partial copy, pseudogene ycf1 (Ψycf1) and ndhF. At the IRb/LSC junction, rpl2 and trnH-GUG were located within the distance of 122–124 bp from each other.

Figure 3 Comparison of the LSC, IR and SSC junction positions in the chloroplast genomes of the C. boreale strains.

Genes above the longer box are transcribed in forward direction and genes below the box are transcribed in reverse direction. Ψ indicates a pseudogene.

Repeat analysis

We investigated the distribution of SSRs that were 1–6 bp in length in the C. boreale cp genomes. We recorded a total of 47, 43 and 50 SSR motifs in 121002, IT232531 and IT301358, respectively (Table S5). Mononucleotide repetition was most prevalent in each cp genome, followed by tri-, penta-and tetra-nucleotide repetition (Fig. 4A). We did not detect di-or hexa-nucleotide SSRs. In terms of sequence context, there were more adenine and thymine residues than cytosine and guanine residues (Fig. 4A). Intergenic and intronic regions contained more SSRs than coding regions, with 41, 37 and 42 instances of SSR occurrence in the non-coding regions in the 121002, IT232531 and IT301358 strains, respectively (Fig. 4B). Most of the SSRs were located in the LSC region followed by those in the IR region (Fig. 4C).

Figure 4 Analyses of simple sequence repeats (SSRs) in C. boreale chloroplast genomes.

(A) The frequency of SSRs per sequence type. (B) The frequency of SSRs in intergenic spacer (IGS), coding sequence (CDS), intron and IGS/CDS. IGS/CDS represents SSRs shared in IGS and CDS. (C) The frequency of SSRs in large single copy (LSC), inverted repeat (IR) and small single copy (SSC) regions.

We detected four different types of long dispersed repeats (LDRs), namely forward (F), palindromic (P), reverse (R) and complement (C) repeats, each with a motif length longer than 30 bp. We identified a total of 19 (12F, 5P, 2R), 18 (11F, 7P) and 16 (9F, 5P, 1R, 1C) repeats in the cp genomes of strains 121002, IT232531 and IT301358, respectively (Fig. 5A; Table S6). F and P repeats were more abundant than C and R repeats. Repeat units of 30–34 bp were the most common, whereas repeat units longer than 40 bp occurred less frequently (Fig. 5A). More LDRs were located in non-coding regions (IGS and introns) than in coding regions (Fig. 5B). Among the protein-coding genes, LDRs were detected in the psaA, psaB and ycf2 in all three C. boreale strains (Table S6). Most LDRs were present in LSC region compared to IR and SSC regions, while some LDRs were shared among LSC, IR and SSC regions (Fig. 5C).

Figure 5 Analyses of long dispersed repeats (LDRs) in C. boreale chloroplast genomes.

(A) The frequency of LDRs classified by the length and type of repeat: forward (F), palindromic (P), reverse (R) and complement (C) repeats. (B) The frequency of LDRs in intergenic spacer (IGS), coding sequence (CDS), intron, IGS/CDS and IGS/intron. IGS/CDS represents LDRs shared in IGS and CDS. IGS/intron represents LDRs shared in IGS and intron. (C) The frequency of LDRs in different genomic regions.

Phylogenetic analysis

The phylogenetic trees were constructed based on complete cp genome sequences and 77 protein-coding genes that were common to the three C. boreale strains, the 17 other species of the tribe Anthemideae (Asteroideae, Astereaceae), and the outgroup species, L. sativa (Cichorieae, Cichorioideae, Asteraceae). The multiple alignment of complete cp genomes contained 158,397 nucleotide sites in which 11,844 were variable and 3,371 were parsimony informative. The multiple alignment of protein-coding sequences possessed 62,965 nucleotide sites in which 3,065 were variable and 900 were parsimony informative. In each cp sequences, both ML and BI trees revealed similar topologies but minor difference within Chrysanthemum species (Fig. 6; Fig. S4). Two datasets also resulted in similar phylogenetic relationship. All Chrysanthemum sequences were grouped into a single clade together with Opisthopappus taihangensis (Ling) C.Shih with high bootstrap support and Bayesian inference (Fig. 6). Three C. boreale strains were all non-monophyletic, which was also observed in two C. morifolium analyzed. Additionally, Artemisia species were clustered into two clades. Among them, seven species formed a monophyletic group, and other three were located in another clade and were closer to the Chrysanthemum clade.

Figure 6 Cladograms using the maximum likelihood (ML) and Bayesian inference (BI) methods.

(A) ML tree based on the sequences of 77 shared protein-coding genes. (B) BI tree based on the sequences of 77 shared protein-coding genes. (C) ML tree based on the complete chloroplast genomes. (D) BI tree based on the complete chloroplast genomes. Numbers above the branches indicate bootstrap support values in ML trees and BI posterior probability in BI trees.

Discussion

The overall cp genome structures and sequences in the three C. boreale strains examined here were conserved and displayed the classical quadripartite structure of land plant cp genomes (Shen et al., 2018). The gene content, gene order and gene orientation in the cp genomes were conserved. Genomic rearrangements such as inversion of the SSC or of individual genes is common in cp genomes (Liu et al., 2018a). However, there were no definitive genomic rearrangements or gene inversions in the three C. boreale strains. The length differences of cp genomes were observed among strains, which was due to variation mainly in the lengths of the LSC and SSC regions. The IR region, which influences the evolution of cp genomes due to its expansion, contraction, or complete loss (Wicke et al., 2011; Zhu et al., 2016), was similar in length, with only a 1–4 bp difference among strains. Our results are consistent with similar studies of the cotton genus (Gossypium), in which the length of LSC regions accounted for the cp genome size difference (Chen et al., 2017). This is different from studies in duckweed species (Lemnoideae), in which differences in cp genome size were due to differences in IR regions (Ding et al., 2017).

Sequence identity plot and the analyses of SNPs and indels revealed the variable regions in the cp genome of C. boreale. In line with observations in other plant species, the LSC and SSC regions were more divergent than the IR regions, and non-coding regions were more variable than coding regions (Meng et al., 2019; Wang et al., 2018). Among the variable regions in the C. boreale cp genome, the introns of trnK-UUU and ndhA, and the spacers of ndhC_trnV-UAC, ycf1_rps15, trnL-UAG_rpl32, and psbE_petL as well as the coding regions of ycf1 and rpoC2 contained many polymorphisms, suggesting rapid genome evolution due to higher mutation rates than other regions. The trnK-UUU intron was longer than 2.5 kb and encompassed matK, which included six SNPs. This region has been extensively used as a molecular marker for phylogenetic and evolutionary studies (Hausner et al., 2006). Therefore, future studies investigating phylogeny and evolution in relatives of C. boreale are likely to find its cp genome a useful resource.

We also detected variation in the number and distribution of two types of repeats, SSRs and LDRs, in both non-coding (IGS and intron) and coding regions. The occurrence of repeats was more prevalent in the non-coding regions than in the coding regions, similar to reports in other species (Kim et al., 2015a; Meng et al., 2019; Shen et al., 2018). Differential distribution of these repeats is associated with cp genome rearrangement and nucleotide substitution (Weng et al., 2014). Therefore, these repeats could be used to develop genetic markers for phylogenetic studies. The obtained SSR repeats, together with the variable regions could be used to examine the genetic structure, diversity, phylogeny, and differentiation of Chrysanthemum and other Asteraceae species.

The phylogenetic analysis revealed the evolutionary relationships of species in the Anthemideae tribe. The investigated species were clustered into a monophyletic group and were largely classified into two groups: Chrysanthemum and Artemisia. The Chrysanthemum clade included Chrysanthemum species, O. taihangensis, Crossostephium chinense Makino, and unexpected three Artemisia species (A. annua, A. fukudo and A.maritima), while the Artemisia clade included the remaining seven Artemisia species, which was consistent with the previous analysis (Gu et al., 2019). However, other phylogenetic studies showed that all Artemisia species were clustered together and separated from its sister genus Chrysanthemum (Meng et al., 2019; Shahzadi et al., 2020). In their analyses, the three Artemisia species closer to Chrysanthemum in our study formed a separated clade within the Artemisia genus. At least, it is clear that Artemisia species are classified into two groups based on cp sequences but their relationship with Chrysanthemum needs to be further addressed.

Within the Chrysanthemum clade, C. boreale strains were placed in separate branches. Two C. morifolium cultivars from Korea and China were also placed in separate branches. This was similar to an earlier phylogenetic analysis of more diverse Chrysanthemum species that used seven cp regions and a single copy nuclear gene (the chrysanthemyl diphosphate synthase, CDS gene): different strains of C. indicum were located in different branches (Liu et al., 2012). On the other hand, we assembled the nuclear genomic regions encompassing the rRNAs and the nuclear ribosomal internal transcribed spacer (nrITS) of around 5.8 kb in size for three C. boreale strains and C. morifolium cv. Baekma with the same approach for cp genome (Supplemental Data 1). Their phylogenetic relationships based on nuclear sequences indicated that all Chrysanthemum sequences formed a monophyletic group in which C. boreale strain IT301358 was clustered together with C. morifolium cv. Baekma (Fig. S5). These results suggest the close affinity within the Chrysanthemum genus and therefore the classification or circumscription using cp and nrITS sequences would be difficult within Chrysanthemum. Divergence and speciation in the Chrysanthemum genus were suggested to be affected by geographical and ecological factors (Liu et al., 2012). Further research including other cultivars and varieties from different regions, and molecular markers from nucleus genome sequences, may reveal the origin of cultivated chrysanthemum and the genetic relationships within the Chrysanthemum group.

Opisthopappus taihangensis is a monotypic species in the genus (Gu et al., 2019) and its phylogenetic position as a sister taxon of C. boreale was inconsistent with previous studies in which O. taihangensis was basal to the Chrysanthemum group when nrITS sequences were used (Zhao et al., 2010). Considering that nrITS sequences can be as short as 447 bp (Zhao et al., 2010), we would expect fewer informative polymorphisms from ITS than the cp as a whole. However, we cannot exclude the possibility that the phylogeny based on the cp genome was sometimes unreliable due to the mode of inheritance of cp genome (Folk, Mandel & Freudenstein, 2017; Tonti-Filippini et al., 2017). Hybridization between distant species (or relatives) and the subsequent chloroplast capture have also been suggested to underlie discrepancies between the nuclear and cp genomes and consequently cause differences in phylogenetic analysis. Phylogenetic trees based on cp and nuclear data also showed the incongruence within Chrysanthemum as discussed above.

Conclusions

Using next-generation sequencing technology, we compared the complete cp genomes of three C. boreale strains. The gene content, gene order and GC content of all the three cp genomes were conserved. The rapidly evolving divergent regions and repeats we identified could potentially serve as molecular markers in phylogenetic studies. Phylogenetic analyses using other Chrysanthemum species and other species within Anthemideae strongly supported the taxonomic status of the strains within the tribe. The data presented here provide insights into the evolutionary relationships among C. boreale strains and other Chrysanthemum species, and will act as a valuable resource for their molecular identification and breeding, as well as for further biological discoveries.

Supplemental Information

Supplemental Information 1 Characteristics of three Chrysanthemum boreale strains.

(A) The collection areas are marked as “a” for 121002, “b” for IT232531 and “c” for IT301358. (B) The morphology of flower head, ray floret and leaf. The ruler scale is in mm.

Click here for additional data file.

Supplemental Information 2 Alignment of nucleotide sequences of the ycf1 gene.

Only the inconsistent regions between two assembly processes are shown. The number in bp on the left indicates the position of first nucleotide displayed in the coding sequences of ycf1.

Click here for additional data file.

Supplemental Information 3 Alignment of nucleotide sequences at the intron of trnK-UUU.

Only the most divergent regions are shown. The number in bp on the left indicates the position of nucleotide in the complete chloroplast genome.

Click here for additional data file.

Supplemental Information 4 Phylogenetic analyses using the maximum likelihood (ML) and Bayesian inference (BI) methods.

(A) ML tree based on the sequences of 77 shared protein-coding genes. (B) BI tree based on the sequences of 77 shared protein-coding genes. (C) ML tree based on the complete chloroplast genomes. (D) BI tree based on the complete chloroplast genomes. Numbers above the branches indicate bootstrap support values in ML trees and BI posterior probability in BI trees. The scale bars indicate the number of nucleotide substitutions per site.

Click here for additional data file.

Supplemental Information 5 Maximum likelihood tree based on nrITS sequences.

Numbers above the branches indicate bootstrap support values. The scale bars indicate the number of nucleotide substitutions per site.

Click here for additional data file.

Supplemental Information 6 List of chloroplast genomes used for phylogenetic analysis in this study.

Click here for additional data file.

Supplemental Information 7 The best-fit models for phylogenetic analyses of different chloroplast datasets.

Click here for additional data file.

Supplemental Information 8 Distribution of SNPs in the chloroplast genome of C. boreale.

Click here for additional data file.

Supplemental Information 9 Distribution of indels in the chloroplast genome of C. boreale..

Click here for additional data file.

Supplemental Information 10 Distribution of SSRs in the chloroplast genome of C. boreale.

Click here for additional data file.

Supplemental Information 11 Distribution of long dispersed repeats in the chloroplast genome of C. boreale.

Click here for additional data file.

Supplemental Information 12 The nucleotide sequences of the ribosomal RNAs and the nuclear ribosomal internal transcribed spacer.

Sequences from three C. boreale strains and C. morifolium cv. Baekma were provided.

Click here for additional data file.

Additional Information and Declarations

Competing Interests

Author Contributions

DNA Deposition

Data Availability

The authors declare that they have no competing interests.

Swati Tyagi conceived and designed the experiments, performed the experiments, analyzed the data, prepared figures and/or tables, authored or reviewed drafts of the paper, and approved the final draft.

Jae-A Jung performed the experiments, authored or reviewed drafts of the paper, and approved the final draft.

Jung Sun Kim analyzed the data, authored or reviewed drafts of the paper, and approved the final draft.

So Youn Won conceived and designed the experiments, performed the experiments, analyzed the data, prepared figures and/or tables, authored or reviewed drafts of the paper, and approved the final draft.

The following information was supplied regarding the deposition of DNA sequences:

The chloroplast genome sequences of Chrysanthemum boreale are available at GenBank: MN909052 (strain IT232531) and MN913565 (strain IT301358).

The following information was supplied regarding data availability:

Additional data are available in the Supplemental Figures and Tables, and the nucleotide sequences of nrITS regions in three C. boreale strains and C. morifolium cv. Baekma are available in a Supplemental File.

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
