# Peer review of "A comparative analysis of the complete chloroplast genomes of three Chrysanthemum boreale strains"

_PeerJ, doi:10.7717/peerj.9448_

## Round 0.1 · original submission · Minor Revisions

All reviewers responded very favorably to your manuscript. They do have several suggestions for imporvement. Please follow them attentively. I look forward to receiving a definitive version of your interesting manuscript!

Reviewer 1 ·

Basic reporting

The reviewed paper reports the results of chloroplast genome sequencing of three Chrysanthemum boreale strains. Complete chloroplast genome sequences were characterized and used for comparative and evolutionary studies. The article is well structured: manuscript include relative background information to the reported results; results are clearly and unambiguously presented and illustrated by appropriate figures and tables; all necessary raw data are provided; in the Discussion section Authors smoothly referred to adequate literature, sufficient in the respect to the number and content; conclusions are generally well supported. The manuscript is characterized by clear and professional English used throughout, with very few imperfections (detailed information on that issue is included in the "General comments" section).

Experimental design

The reviewed paper is an example of an interesting and valuable study with the correct and complementary methodological side. The applied methodology, high-throughput sequencing, provides high-quality data with many applications. Here, authors not only report complete chloroplast genomes of three Chrysanthemum boreale strains, but also based on the molecular data verified the systematic relationships of the studied material.

Validity of the findings

The described results perfectly correspond and are complementary to previous studies on history, evolution, and diversity of genus Chrysanthemum. The observations included in the reviewed manuscript may become a valuable element of the discussion not only in case of studies on evolution and diversity of Chrusanthemum boreale, but also for other closely related taxa.

Additional comments

Here I would like to list my detailed, minor comments concerning the manuscript:

L.22
"regions in Korea" replace with "regions of Korea"

L.64
- what is the difference between "conserved" and "stable", it sounds like redundancy, please rephrase this sentence

L.64-65
"maternally inherited" - it is only partially true, there are a number of examples where chloroplast genome is inherited with pollen grains (paternal inheritance; e.g. in Pinus, Picea), please rephrase it by pointing at rather uniparental mode of cp genome inheritance not maternal.

L.71-74
"Although whole cp sequences, gene content, and gene order are conserved..."
I would recommend to rephrase it a little bit to avoid redundancy:
"Although the structure of cp genome is generally conserved..."

L.95
There is no "Kim et al. 2006" among the references.

L.140
Have you tested also other substitution models? Please add the appropriate information here.

L. 147-148
"We used the previously reported cp genome of strain 121002, which was 151,012 bp (Won et al. 2018)."
Something is missing here, isn't it? Please add adequate information here by giving the reason/purpose why you use that sequence.

L.162
"duplicates" replace with "duplicated"

Figure 1
The picture could be of better quality (higher resolution) - the gene names are not sharp enough.

Figure 2
Low-quality picture - the gene names are hard to read.

Figure 3
I suggest adding the LSC, IR and SSC symbols above the arrows indicating the elements of the cp genome.

Figure 4
I suggest to change the color scheme used in Figure 4A - it is hard to distinguish some of the used colors.

Figure 5
Low-quality picture - the bootstrap values supporting the tree nodes are not clear.

Table 1.
I suggest replacing "Number of genes" with "Number of unique genes"

Table 2.
In the main text you have mentioned that there are 15 genes containing introns, but in the table the asterisk (*) is placed next to only 14 genes (rpl2, rps16, rpoC1, trnA-UGC, trnI-GAU, trnK-UUU, trnL-UAA, trnS-CGA, trnV-UAC, ndhA, ndhB, atpF, clpP, ycf3).

Reviewer 2 ·

Basic reporting

This manuscript may have the potential to contribute to our understanding of relationships of geographical species, such as Chrysanthemum boreale strains, using plastome phylogenomics. While the current version is clear enough in some section for revision, other parts of the manuscript have some issues that make very difficult to understand it.

There is no information of where the tree, alignments and other appropriate data was deposited.

Experimental design

This paper focuses in three C. boreale strains varied in morphology and geographic, but you do not provide enough background about these species. What is the current classification for these strains based on morphology? Are they believed to be one species in taxonomic? What is the distribution for these strains separately? Also, you ended up focusing more in describing the three chloroplast and not much attention is put on the phylogenetics results and the systematics of the C. boreale strains, which is I understand is the main point of the paper.

Validity of the findings

It is clear that you generated new valuable data of C. boreale strains. In Hwang et al. (2013) (Karyomorphological Analysis of Wild Chrysanthemum boreale Collected from Four Natural Habitats in Korea), this species was identified four genotypes, I think generate new data of the other strains could improve the systematics of C. boreale.

On first mention of species name in the paper need to in correct scientific name with author from IPNI (https://www.ipni.org/), such as Line 20, 41, 54, 55, 76 etc.

Additional comments

Abstract:

Line 20: “Chrysanthemum boreale” should be “Chrysanthemum boreale Makino”
Line 29: Check for “80 unique protein-coding genes” or “80 protein-coding genes”

Materials & Methods

Line 135: Here, authors mentioned 73 shared protein-coding sequences, but unclear why not used complete cp genomes, which could improve node support.

Results

Line 199-201: The length of ycf1 protein in 121002 strain almost 640 amino acids shorter than IT232531 and IT 301358. I suggest authors should examine by Sanger sequence or agarose gel electrophoresis.
Line 246-247: “The total length of the cp genomes was within the expected size range of 120–160 kb (Thode & Lohmann 2019)” repeat with above.

References

Scientific name should be in italics.

Reviewer 3 ·

Basic reporting

The overall structure and format of the article is appropriate for the journal. The manuscript by Tyagi and colleagues sequenced the chloroplast genomes of three Chrysanthemum boreale strains. They systematically analyzed three chloroplast genomes by assembled de novo, annotated, compared with one another, and Phylogenetic analysis. The three complete chloroplast genomes will be valuable genetic resources for studying the population genetics and evolutionary relationships of Asteraceae species. However, I have several suggestions as listed below for helping the authors to improve their manuscript:

1. Line 48-50: Chrysanthemum is considered to include around 40 different species native to Eurasia, especially in Korea, China, and Japan (Liu et al. 2012). However, some species and varieties are narrowly distributed in specific habitats (Kondo et al. 2003; Liu et al. 2012).
Line 284-286: Considering that ITS sequences can be short as 447 bp (Zhao et al. 2010), we would expect fewer informative polymorphisms from ITS than the cp as a whole.

They provide information for Chrysanthemum species, however, only very limited evolutionary information is available in the introduction and discussion about their evolutionary. It is suggested to increase the evolutionary relationship based on nuclear genome of Chrysanthemum species.

2. Line 50-53: Of particular importance to the present study is a wild relative, Chrysanthemum boreale, which bears small yellow flowers, and occurs in natural stands in eastern Asia (Hwang et al. 2013; Kim et al. 2014). Comparative transcriptomic analysis revealed that C. boreale diverged from the commercial cultivar Chrysanthemum morifolium about 1.7 million years ago.

I believe that the Chrysanthemum boreale of the article has important application value. However, the materials introduction to the research materials of Chrysanthemum boreale, which bears small yellow flowers, and occurs in natural stands in eastern Asia, is very limited. If the three Chrysanthemum boreale strains have no representative significance and value, this paper has very limited theoretical and practical significance.

Experimental design

The research methods are reasonable and feasible.

Validity of the findings

Statistical analysis needs to be added to make conclusions. The resolution Figure 2 and Figure 5 need to be further improved.

Additional comments

The article should be accepted if the authors make some revisions, minor enough that I would NOT necessarily need to re-review it.

---

## Round 0.2 · Minor Revisions

Please address the few final points highlighted by reviewer #1

Reviewer 1 ·

Basic reporting

The reviewed manuscript is the resubmission of a paper which reported the results of chloroplast genome sequencing of three Chrysanthemum boreale strains.
Complete chloroplast genome sequences were characterized and used for comparative and evolutionary studies. The article is well structured: manuscript include relative background information to the reported results; results are clearly and unambiguously presented and illustrated by appropriate figures and tables; all necessary raw data are provided; in the Discussion section Authors smoothly referred to adequate literature, sufficient in the respect to the number and content; conclusions are generally well supported. The manuscript is characterized by clear English used throughout, with very few imperfections (detailed information on that issue is included in the "General comments" section.

Experimental design

The reviewed paper is an example of an interesting and valuable study with the correct and complementary methodological side. The applied methodology, high-throughput sequencing, provides high-quality data with many applications. Here, authors not only report complete chloroplast genomes of three Chrysanthemum boreale strains, but also based on the molecular data verified the systematic relationships of the studied material.

Validity of the findings

The described results correspond and are complementary to previous studies on history, evolution, and diversity of genus Chrysanthemum. The observations included in the reviewed manuscript may become a valuable element of the discussion not only in case of studies on evolution and diversity of Chrusanthemum boreale, but also for other closely related taxa.

Additional comments

Thank you very much for your effort made to improve the manuscript. I am satisfied with the answers received for my previous comments and current version of the text.

However I have noticed some minor drawbacks which needs improvements:

L.55
"Almost 8 species" - I believe that this is a unfortunate expression here, authors should make this statement more accurate i.e. whether it is 8 or more/less species.

L.235
"ndfF" should be replaced with "ndhF"

L.235-236
"At the IRb/LSC junction, rpl2 and trnH-GUG were located with the distance of 122 bp to 124 bp." should be rephrased "At the IRb/LSC junction, rpl2 and trnH-GUG were located within the distance of 122 bp to 124 bp from each other"

L.260
"The phylogenetic trees were constructed with complete cp genome..." replace with "The phylogenetic trees were constructed based on complete cp genome sequences..."

L.350-351
"Phylogenetic trees using the cp and nuclear data..." should be replaced with "Phylogenetic trees based on cp and nuclear data..."

Table 1
I do not understand why there is information that the number of genes with introns is 13-15 depending on the C. boreale strain, whereas in line 185 you wrote that there is 16 such genes. Additionally, Table 3 contains details on intron content in 16 chloroplast genes in C. boreale strains.

Reviewer 2 ·

Basic reporting

no comment

Experimental design

no comment

Validity of the findings

no comment

Reviewer 3 ·

Basic reporting

The overall structure and format of the article is appropriate for the journal.

Experimental design

This original primary research is within the scope of the journal.

Validity of the findings

Conclusions are well stated.

Additional comments

The article meets the PeerJ criteria and should be accepted as is.

---

## Round 0.3 · accepted · Accept

Thank you for addressing the few final issues!